# Should We Use the Men Load–Velocity Profile for Women in Deadlift and Hip Thrust?

**DOI:** 10.3390/ijerph20064888

**Published:** 2023-03-10

**Authors:** Raúl Nieto-Acevedo, Blanca Romero-Moraleda, Almudena Montalvo-Pérez, Agustín Valdés-Álvarez, Carlos García-Sánchez, Daniel Mon-López

**Affiliations:** 1Deporte y Entrenamiento Research Group, Departamento de Deportes, Facultad de Ciencias de la Actividad Física y del Deporte (INEF), Universidad Politécnica de Madrid, C. de Martín Fierro 7, 28040 Madrid, Spain; 2Department of Physical Education, Sport and Human Movement, Universidad Autónoma de Madrid, 28049 Madrid, Spain; 3Faculty of Sport Sciences, Universidad Europea de Madrid, 28670 Madrid, Spain; 4LFE Research Group, Department of Health and Human Performance, Faculty of Physical Activity and Sport Science (INEF), Universidad Politécnica de Madrid (UPM), 28040 Madrid, Spain

**Keywords:** injuries, hamstrings, sex differences, velocity-based training

## Abstract

Injuries are common in team sports and can impact both team and individual performance. In particular, hamstring strain injuries are some of the most common injuries. Furthermore, hamstring injury ratios, in number of injuries and total absence days, have doubled in the last 21 seasons in professional soccer. Weakness in hip extensor strength has been identified as a risk factor in elite-level sprinters. In addition, strength imbalances of the hamstring muscle group seem to be a common cause of hamstring strain injuries. In this regard, velocity-based training has been proposed to analyze deficits in the force–velocity profile. Previous studies have shown differences between men and women, since there are biomechanical and neuromuscular differences in the lower limbs between sexes. Therefore, the aim of this study was to compare the load–velocity profile between males and females during two of the most important hip extension exercises: the hip thrust and the deadlift. Sixteen men and sixteen women were measured in an incremental loading test following standard procedures for the hip thrust and deadlift exercises. Pearson’s correlation (*r*) was used to measure the strength of the correlation between movement velocity and load (%1RM). The differences in the load–velocity relationship between the men and the women were assessed using a 2 (sex) × 15 (load) repeated-measures ANOVA. The main findings revealed that: (I) the load–velocity relationship was always strong and linear in both exercises (*R*^2^ range: 0.88–0.94), (II) men showed higher velocities for light loads (30–50%1RM; effect size: 0.9–0.96) than women for the deadlift, but no significant differences were found for the hip thrust. Based on the results of this study, the load–velocity equations seem to be sex-specific. Therefore, we suggest that using sex-specific equations to analyze deficits in the force–velocity profile would be more effective to control intensity in the deadlift exercise.

## 1. Introduction

Injuries are common in team sports such as soccer [1] and rugby [2]. Incidence rates in these sports can involve a significant financial cost for their sporting organizations [3,4,5]. Additionally, injuries can impact team and individual performance [1,6,7,8], as well as physical and psychological well-being [9]. Hamstring-strain injuries (HSIs) are some of the most common injuries in athletes [10,11,12]. Furthermore, hamstring injury ratios—in number of injuries and total absence days—have doubled in the last 21 seasons in professional soccer [13]. The majority of hamstring muscle injuries occur during non-contact situations, classified as ‘indirect muscle injuries’ or ‘muscle strains’. They usually occur when the applied force exceeds the capacity of the tissue [14].

According to Croisier et al. [15] HSIs have been associated with hamstring strength. Previously injured hamstrings expressed 30% deficits in eccentric force development rate upon return to sport after HSI [16]. Moreover, in previous studies, muscle strength performance disorders were isokinetically detected in about 70% of cases after hamstring strain [17]. Despite the difficulties in studying HSIs, it is important to understand why they occur. It appears that risk factors are a complex network of determinants. Strength imbalances of the hamstring muscle group seem to be a common cause of HSI [18]. In a recent review [18], the authors considers as a strength imbalance knee flexor weakness, bilateral knee flexor strength asymmetry and low ratios of knee flexor to knee extensor strength, otherwise known as hamstrings to quadriceps (H:Q) ratios. Decreased hamstring strength relative to the quadriceps is implicated as a potential mechanism for increased lower extremity injuries with H:Q deficits [19]. Athletes with an H:Q deficit tend to stabilize the knee joint by primarily using the quadriceps muscles, and this deficit can alter the co-contraction of the hamstrings and quadriceps [19]. Women appear to preferentially use the quadriceps more than males in order to stiffen and stabilize the knee joint [20]. In addition, a weakness in hip extension strength was identified as a prospective risk factor for HSI in elite-level sprinters [16]. A recent study shows that elite sprinters present higher levels of maximum and relative strength, absolute and relative peak force and lower levels of strength deficit [21]. Given the limited time available to produce force during a sprint, the main manifestation of force is explosiveness, defined as the development of maximal force in minimal time or rate of force development (RFD) [22,23]. Neuromuscular function is crucial to sprint performance because the activity and the interaction of the central nervous system with the muscles ultimately influence muscle RFD [23]. The RFD refers to the relationship between time and the force applied [24]. A simple alternative of RFD is the force–velocity (FV) profile. FV is a way to assess an athlete’s force and velocity production capabilities during ballistic tasks such as jumping and sprinting [25]. FV profiles are being increasingly used to identify areas of potential improvement [26].

In this regard, FV profiles can be generated using different way of measuring force. By creating an FV profile, we can identify the potential areas for improvements [27]. Velocity-Based Training (VBT) has been proposed as an accurate method to monitor and prescribe resistance training intensities and volumes. VBT uses the velocity of the bar to determine the relative load (%1RM) [28]. Recent studies have shown that the FV relationship is stable when using the velocity of the bar [29]. The two velocity variables most commonly used in practice and scientific research are mean velocity (MV) (i.e., the average velocity across the entire concentric phase) and peak velocity (PV) (i.e., the maximum instantaneous velocity reached during the concentric phase). However, mean propulsive velocity (MPV) (i.e., the average velocity from the start of the concentric phase until the acceleration is less than gravity [30,31] has also been proposed as an alternative [32]. Interestingly, several studies have found a nearly perfect association between MPV and percentage of 1 Repetition Maximum (%1RM) in different exercises, including squat, half squat [33] and leg press [34]. Most of the studies mentioned above analyzed the load–velocity profile in men, which could be considered as a limitation, since previous studies have shown that men show higher velocity values at different %1RMs than women in the bench press, squat, inclined bench press and seated military press exercises [30,35,36]. These results suggest that the load–velocity relationship seems to be sex-specific. In addition, evidence supports that employing different strategies for men and women may be more effective at improving power [37]. In fact, Antunes et al. [38] observed that men produced higher power in both absolute and relative terms when examining the power output values of elite weightlifters during competition.

Furthermore, the deadlift and the hip thrust are two exercises that improve the posterior muscles of the legs, and they are commonly used for training power in elite athletes [39]; the deadlift and the hip thrust involved key muscle groups for acceleration [40]. Therefore, the purpose of the current study was to determine if there are differences in mean propulsive velocity (MPV) from 30 to 100%1RM in the hip thrust and deadlift exercises between genders. It was hypothesized that men would have higher velocities in each %1RM than women in both the hip thrust and the deadlift.

## 2. Materials and Methods

### 2.1. Participants

Although the power analysis conducted in previous studies revealed that sample sizes of only 3–9 participants were needed to detect the differences in mechanical variables (force, velocity and power) [24], we conservatively recruited sixteen men (age = 25.63 ± 3.79 years; body mass = 75.79 ± 8.64 kg; height = 175.81 ± 7.34 cm) and sixteen women (age = 25.06 ± 5.37 years; body mass = 61.94 ± 4.25 kg; height = 165.06 ± 5.72 cm) who participated voluntarily. Inclusion criteria were (1) having at least one year of resistance training experience in hip thrust and deadlift; and (2) not having any health or musculoskeletal injuries that could compromise the testing. After being informed of the purpose and testing procedures, subjects signed a written informed consent form prior to participation. The present investigation was approved by the Research Ethics Committee of the Universidad Politécnica de Madrid and was conducted in accordance with the Declaration of Helsinki [41].

### 2.2. Experimental Design

Participants underwent a preliminary session in which they were familiarized with the testing equipment and the exercise protocol. This session was also used for body composition assessment, personal data and health history questionnaire administration. Participants came to the laboratory on two more occasions separated by 48–72 h. Each exercise was tested on each occasion. Individual load–velocity profiles were determined by means of an incremental loading test that followed standard procedures for the hip thrust (HT) [42] and deadlift (DL) [43] exercises. The test sessions were conducted at the same place and time of day (±1 h) for each subject and under the same environmental conditions. Participants were requested to avoid strenuous exercises and beverages containing caffeine/alcohol for 24 h prior to testing on both of these sessions.

### 2.3. Testing Procedures

The following considerations were taken into account for the deadlift technique. The subject had to lift the bar while avoiding countermovement of the hips, ending with arms and legs completely extended. A self-selected width with a mixed grip (one arm pronated and one arm supinated) was used. It was performed starting from the floor and at the same height for all participants (approximately the average distance between the knee and the ankle), with a stance approximately shoulder-width apart and with both feet positioned flat on the floor in parallel or slightly externally rotated, while keeping a neutral spine, chest up and head in line with the spine. The subjects were then instructed to pull the bar in a vertical direction at the maximum intended velocity until their body was fully erect and were instructed to maintain the final static position for ~1 s [43].

The technique for the hip thrust exercise was as follows: subjects had their upper backs positioned on a bench (the lower angle of the scapula at the end of the bench); feet were slightly wider than shoulder-width apart, with toes pointed forward or slightly outward. The barbell was padded with a thick bar pad and placed over the subjects’ hips, and they were instructed to thrust the bar upward while maintaining a neutral spine and pelvis [42].

The warm-up protocol consisted of 3 min of stationary cycling at a self-selected easy pace and 5 min of joint mobilization exercises, followed by 6 repetitions with fixed loads of 30 and 20 kg for men and women, respectively, for both exercises. Individual load–velocity relationships and 1RM strength were determined using a progressive loading test. MPV was tested due to strong correlations observed between mean propulsive velocity (MPV) and load (%1RM) in previous studies [44,45]. The initial load was set at 20 kg and gradually increased using MPV as a parameter for adding load [35]. The next protocol was the following: initially, in increments of 20 kg until an MPV of 0.8 m·s^−1^ was reached, 3 repetitions were performed. Two repetitions were performed when the MPV was between 0.8 and 0.6 m·s^−1^ (10 kg increments), and only one repetition from the present until the end of the test. Increments of 5 kg were used when the MPV ranged from 0.6 to 0.5 m·s^−1^, and 2.5 kg increments were used when the MPV was less than 0.5 m·s^−1^ to 1RM. The heaviest load that each subject could properly lift while completing a full range of motion and without any external help was considered to be the 1RM. Inter-set rests were fixed at 3 min to reduce possible neural or mechanical fatigue [35]. Only the best repetition (fastest and executed correctly) at each load was considered for subsequent analysis. All repetitions were recorded with a linear velocity transducer (Speed4Lifts, v2.0, Madrid, Spain), which has been previously validated [46]. Strong verbal encouragement (e.g., “let’s go”, “keep going”) was provided during all tests to motivate participants to give maximal effort.

### 2.4. Statistical Analysis

Data are presented as means (M) and standard deviation (SD), standard error of the estimate (SEE) and Pearson’s correlation coefficient (r). The normal distribution of the data was confirmed by Shapiro–Wilk, and the homogeneity of variances was confirmed by Levene’s test (*p* > 0.05). Linear regression and Pearson’s correlation coefficient were used to check the relationship between measured and predicted MPV values. The standard error of the estimate (SEE) was calculated as the residual’s standard deviation of the variation around the regression line. ANOVA was applied to compare each dependent variable (i.e., mean velocity values attained at each %1RM, 1RM strength, and mean test velocity), with sex (men and women) as the between-participant factor. When significant differences were observed, a Bonferroni ’s post hoc comparison was performed. The effect size of the differences in the 1RM strength and in the velocity of the 1RM was compared between sexes (men vs. women) through the Cohen’s effect size (ES). The criteria for interpreting the magnitude of the ES were trivial (<0.2), small (0.2–0.6), moderate (0.6–1.2), large (1.2–2.0) and extremely large (>2.0) [47]. Significance level was set at *p* < 0.05. Analyses were carried out using a custom spreadsheet (Microsoft Excel version 16.69.1) and JASP software version 0.16.4 (Nieuwe Achtergracht, Amsterdan).

## 3. Results

### 3.1. 1RM Strength

A significant difference was found between sexes for the 1RM load in HT (men: 179.8 ± 25.9 kg, women: 122.3 ± 22.1 kg) (*p* < 0.01; ES = 2.27) and DL (men: 146.8 ± 20.5 kg and women: 94.8 ± 13.1 kg) (*p* < 0.01; ES = 2.90).

The second-degree polynomial equation obtained from the relationship between relative load (%1RM) and MPV is represented in Figure 1. A very strong association between these two variables could be observed for the hip thrust (R^2^ = 0.88) and the deadlift (R^2^ = 0.94) for both sexes. The MPV associated with each %1RM was obtained from these polynomial fits, from 30%1RM onwards, in 5% increments (Table 1). Mean MPV values of all subjects are shown in Figure 2.

### 3.2. Comparison of the Load–Velocity Relationship between Sexes in Hip Thrust and Deadlift

A significant sex–load interaction was observed for DL (*p* < 0.01; ES = 0.6). However, no significant sex–load interaction was observed for HT (*p* = 1.00). The ANOVA test applied to the mean velocity attained at each test %1RM revealed that men achieved statistically higher values than women (*p*-range: <0.001–0.05). These differences between sexes were observed in light loads (30–55%1RM; ES = 0.9–0.96) in the DL exercise but not in the HT exercise (*p* > 0.05). Men showed significant higher mean MPV for all loads compared to women for both DL and HT (*p* < 0.01; ES = 0.68–0.54, respectively).

### 3.3. Predicting Load (%1RM) from Velocity Data in the Hip Thrust and Deadlift

The prediction equation used in previous studies [35,44] was used to estimate the %1RM in hip thrust from mean propulsive velocity and is as follows:MEN: Load (%1RM) = 8 × 10^−6^ MPV^2^ − 0.0101 MPV + 1.2335 (R² = 0.894; N = 16; SEE = 0.078)
WOMEN: Load (%1RM) = −9 × 10^−6^ MPV^2^ − 000078 MPV + 1.1003 (R² = 0.880; N = 16; SEE = 0.077)

The prediction equation to estimate the %1RM in deadlift from mean propulsive velocity is as follows:MEN: Load (%1RM) = 2 × 10^−5^ MPV^2^ − 000163 MPV + 1.6687 (R² = 0.938; N = 16; SEE = 0.086)
WOMEN: Load (%1RM) = −6 × 10^−6^ MPV^2^ − 0.0109 MPV + 1.3812 (R² = 0.947; N = 16; SEE = 0.065)

## 4. Discussion

The aim of this study was to compare the load–velocity profile between males and females during the hip thrust and the deadlift. The main finding of this study was that women have significantly lower velocity values at light loads (<55%1RM) when compared to men in the deadlift. Furthermore, the absolute load for both exercises (hip thrust and deadlift) was significantly higher for men. Another major finding was that men have significantly higher velocities on average for all loads. Mean velocities attained with each %1RM are very similar to those reported in previous research on the DL exercise [43,48,49] and HT exercise [42].

To the best of our knowledge, the present study is the first to examine the relationship between relative load (%1RM) and mean velocity between men and women for the hip thrust and deadlift simultaneously. Our results are in line with those obtained by Pareja-Blanco et al. [35], García-Ramos et al. [30] and Balsalobre-Fernández et al. [36], who found sex differences from light to moderate loads for the bench press, inclined bench press, military press and squat. These previous studies showed significant differences in favor of men until 85–90%1RM, whereas we only found significant differences under 55%1RM for the deadlift and none for the hip thrust.

Although the exercise protocols used do not seem to be much different, it is difficult to explain why we did not obtain similar results; however, it might be attributable to the use of free-weight exercises instead of the Smith machine. Concerning sex differences, another reason could be that women have a higher proportion of slow muscle fibers when compared to men [50]. Nevertheless, the underlying mechanisms need to be studied further. Furthermore, differences in fat-free mass (FFM) between sexes may partly explain why women produced significantly higher velocities when normalized to FFM [37]. Despite this, men had significantly higher skeletal muscle mass than women in both absolute terms and relative to body mass (38% vs. 31%) [51]. This could be one reason why men have higher velocities than women at the same relative load—they have more muscle mass to move this load. Moreover, these authors indicated that power output appears to be related to greater muscle mass and strength since they observed a significant correlation between FFM and strength (1RM). Regarding power, Margaret et al. [37] reported that absolute peak velocity was only higher in men at lower intensities (i.e., 30%, 60%1RM). Furthermore, Thomas et al. [52] found significantly higher power outputs in men in the squat jump and bench press throw in a range of loads (30% to 70%1RM). Additionally, the optimal load for all the exercises tested occurred in the range of 30–60%1RM [52]. Although the identification of the optimal load was not the purpose of the current study, these studies support the importance of our results because we found differences with moderate loads (30 to 55%1RM).

Another possible reason for the differences between genders could be the height and tested impact on range of motion (ROM). Variations in the ROM of the concentric phase influences several biomechanical factors and can affect the development of force, rate of force development, and activation and synchronization of motor units [53]. Martínez-Cava et al. [54] showed how the ROM influences MPV. They found that ROM affected the 1RM strength, load-velocity profiles and the contribution of the propulsive phase. This could partly explain the gender differences in MPV due to the leg length differences between men and women.

These results need to be interpreted with caution because we did not evaluate the differences in the load–velocity profile between strong and weak participants separately for each sex. This may have been another factor that influenced the results. However, Torrejón et al. [55] found that the load–velocity profile differed more between men and women than between individuals with different strength levels. This data suggest that the differences between men and women are not directly caused by their different strength levels even when divided men and women into two subgroups of strong and weak participants according to their 1RM in relation to body mass.

These findings have important practical applications: (I) one can determine the %1RM that is being used through the MPV with any given load; (II) strength can be estimated from movement velocity and submaximal loads, and hence the potential increased injury risk from the standard 1RM protocol can be avoided [56,57]; (III) one can use sex-specific equations to create force–velocity profiles and identify the potential areas for improvements. Consequently, these practical applications could allow for the detection of FV profile deficits in athletes with previous hamstring injuries who have shown a 30% deficit in eccentric force development rate [16].

## 5. Conclusions

This study suggests that men produce significantly higher velocity values across loads <55%1RM for the deadlift but not for the HT exercise. In addition, a general equation may be used to predict relative load (%1RM) in men and women, but in order to improve the prediction values, a sex-specific equation is recommended.

Our results make it possible to determine the real effort experienced by both sexes when training with loads from 30 to 55%1RM for the deadlift. Nonetheless, it would be interesting if future research compares athletes with similar strength levels or sport-specific training regimens.

## Figures and Tables

**Figure 1 ijerph-20-04888-f001:**
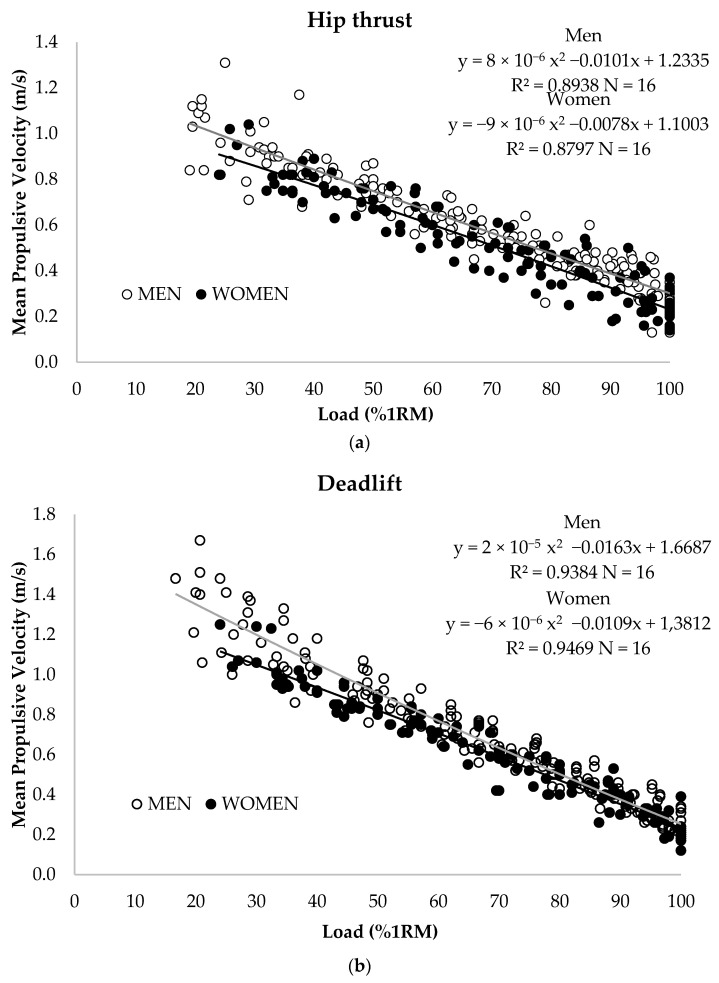
Relationship between relative load (%1RM) and mean propulsive velocity (MPV) for women (filled dots and solid line) and men (open dots and dashed line) in hip thrust (**a**) and deadlift (**b**). R^2^ = Pearson’s multivariate coefficient of determination. N = number of trials included in the regression analysis.

**Figure 2 ijerph-20-04888-f002:**
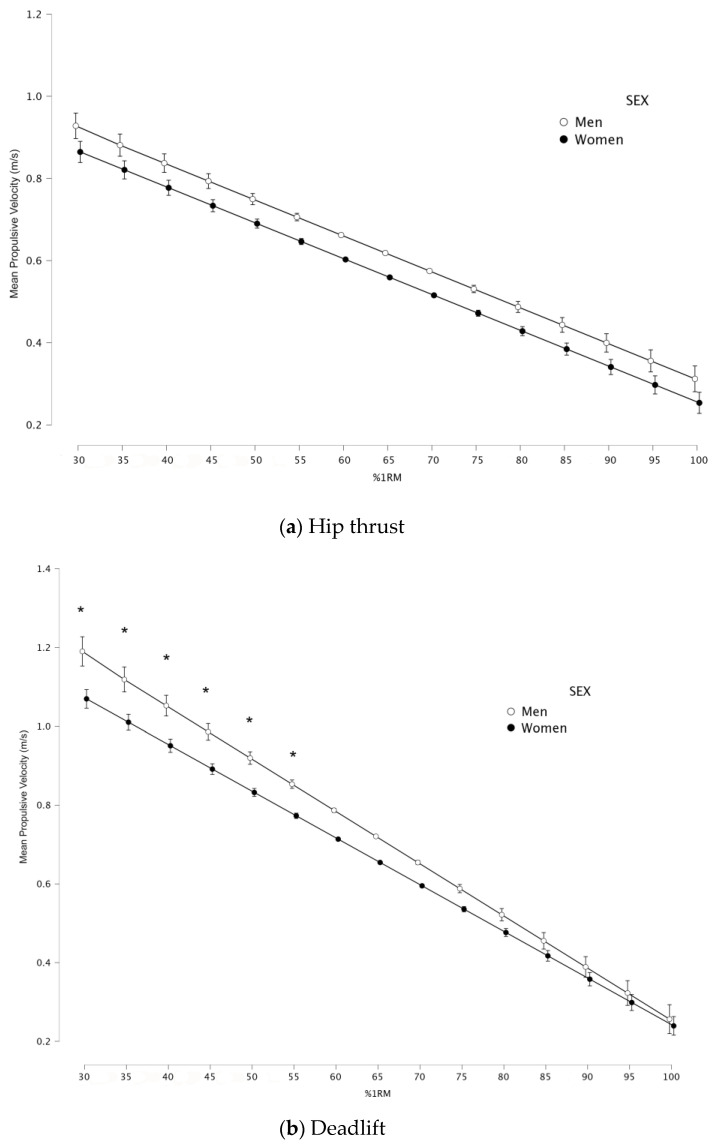
Relationship between relative load (%1RM) and mean propulsive velocity (MPV) for women (filled dots and solid line) and men (open dots and dashed line) in deadlift (**a**) and hip thrust (**b**) (data averaged across the participants). Significant differences between the sexes * *p* < 0.05.

**Table 1 ijerph-20-04888-t001:** Estimated mean propulsive velocity values for each %1RM in the hip thrust and deadlift exercises for men (*n* = 16) and women (*n* = 16) derived from the individual load–velocity relationships.

Load (%1RM)	Hip Thrust		Deadlift	
	Men	Women	Men	Women
30	0.93 ± 0.10	0.86 ± 0.07	1.19 ± 0.13 ***	1.07 ± 0.07
35	0.88 ± 0.09	0.82 ± 0.07	1.12 ± 0.12 **	1.01 ± 0.07
40	0.84 ± 0.08	0.78 ± 0.07	1.05 ± 0.11 **	0.95 ± 0.06
45	0.79 ± 0.08	0.73 ± 0.6	0.99 ± 0.10 *	0.89 ± 0.06
50	0.75 ± 0.07	0.69 ± 0.06	0.92 ± 0.09 *	0.83 ± 0.06
55	0.71 ± 0.07	0.65 ± 0.06	0.85 ± 0.08	0.77 ± 0.05
60	0.66 ± 0.06	0.60 ± 0.06	0.79 ± 0.08	0.71 ± 0.05
65	0.62 ± 0.06	0.56 ± 0.06	0.72 ± 0.07	0.65 ± 0.05
70	0.58 ± 0.06	0.52 ± 0.06	0.65 ± 0.06	0.60 ± 0.04
75	0.53 ± 0.05	0.47 ± 0.06	0.60 ± 0.06	0.54 ± 0.04
80	0.49 ± 0.05	0.43 ± 0.06	0.52 ± 0.05	0.48 ± 0.04
85	0.44 ± 0.05	0.39 ± 0.06	0.46 ± 0.05	0.42 ± 0.04
90	0.40 ± 0.06	0.34 ± 0.06	0.39 ± 0.05	0.36 ± 0.05
95	0.36 ± 0.06	0.30 ± 0.07	0.32 ± 0.05	0.30 ± 0.05
100	0.31 ± 0.06	0.25 ± 0.07	0.26 ± 0.05	0.24 ± 0.05
Mean	0.72 ± 0.07 **	0.65 ± 0.05	0.62 ± 0.06 **	0.56 ± 0.06

Notes: Values are mean (M) ± standard deviation (SD). All velocity values correspond to the mean propulsive velocity. %1RM: relative load expressed as percentage of one-repetition maximum. Significant differences between the sexes * *p* < 0.05, ** *p* < 0.01, *** *p* < 0.001.

## Data Availability

The data presented in this study are available on request from the corresponding author.

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
