# Peer review of "Should We Use the Men Load–Velocity Profile for Women in Deadlift and Hip Thrust?"

_ijerph, 2023, doi:10.3390/ijerph20064888_

Round 1

Reviewer 1 Report

Overall comment:

The authors present novel data on force-velocity relationships in hip thrust and deadlift, considering gender specific differences.

In the introduction of the manuscript the authors provide a broad overview of force velocity profile and hamstring injuries. Although hamstring maximal strength, especially eccentric, has been reported as a contributing factor to hamstring injury, no information is available linking sub maximal loads and injuries. Therefore, further evidence should be provided to substantiate the link between force velocity profile and injury reduction.

The methods section do not fully describe the statistical procedures used to analyze the data.

The manuscript could benefit from major revisions.

Specific comments:

LINE 17:               please modify “hip extension” to “hip extensors”

LINE 18:               please clarify the relationship between maximal strength and explosive strength, please consider referencing https://doi.org/10.3390/life10110282 for further details regarding force-velocity relationship in sprints.

LINE 46:               “Deficits in explosive strength would limit the ability to perform maximal 46 strength during sprinting” citation needed

LINE 67:               please modify “highly” to “commonly”

LINE 98:               please avoid speaking in first person

LINE 110:             please provide further detail regarding the height at which the participants positioned their upper back on the bench

LINE 119:             how was the height of the bar at the start of the movement kept consistent for the lightest loads? Was it the same as the heavier loads?

LINE 121:             please provide reference regarding the selection of mean propulsive velocity as dependent variable

LINE 130:             what warding was used to provide encouragement?

LINE 149:             please report the results for the two exercises in the same order the exercises were presented in the introduction

LINE 152:             please modify “very close” with an appropriate descriptor for the strength of the relationship

LINE 165, 167:    in Figure 3 (a) and 3 (b) please use superscript for the quadratic terms in the equations

LINE 173, 175:    the readability of Figure 4 (a) and 4 (b) would benefit from the inclusion of a clearly visible title stating the exercise investigated, similar to Figure 3 (a) and 3 (b)

LINE 180:             methods for the procedure have not been described in the statistical analysis paragraph

LINE 180:             please provide further information on how relative strength ratio is reported, as the data presented In Table 2 seem to contradict the data provided for the participants body mass and 1RMs

LINE 182:             please modify “analyse” to “analyze”

LINE 210:             please modify font and size for “relative strength ratio”

LINE 221:             could gender differences in height had an impact on ROM an MCV?

LINE 240:             please rephrase

LINE 245:             again, the inference on injury risk in athletic cohorts  is stretched from the data provided

Reviewer 2 Report

General questions

Congratulations for the nice work. This work presents a good scientific relevance. However, some issues need to be clarified ((methods and results (equation)) in addition to other small corrections.

Major questions

- Line 120: “initially in increments of 20 kg until an MPV of 0.8 m·s-1 was reached, performing 3 repetitions with each load. Two repetitions were performed when the MPV was between 0.8 and 0.6 m·s-1 (10 kg increments), and only one repetition for higher loads. Increments of 5 kg were used when the MPV ranged from 0.6 to 0.5 m·s-1 and 2.5 kg increments were used when the MPV was less than 0.5 m·s-1 to 1RM. The heaviest load that each subject could properly lift while completing a full range of motion and without any external help was considered to be the 1RM”.

Everything is OK. I think the authors could make it clearer that the MPV was used as a parameter for increasing the loads, but the speed was individual for each subject. The way it is written gives the impression that the MPV was a target speed criterion for all subjects, when in fact the speed was individualized. it's not that?

- Line 156: “...for each 1RM”: ...for each %1RM

- Line 187: “... mean test velocity”: why not mean propulsive velocity? RSR is not a measure of speed.

- Line 192: 3.4. Predicting load (%1RM) from velocity data in the hip thrust and deadlift: What is the level of significance of the equations?

- Line 195-204: Isn't it the opposite? Predict speed (MVP) using the formula instead of strength (%1RM)?

- Line 251: I think it's predicting MVP and not strength.

Minor questions

- References: standardize manuscript titles (are the initial letters capitalized or not? standardize); standardize journal names (abbreviated or in full);

Reviewer 3 Report

Dear Authors,

Your work is very interesting and I congratulate the authors by such a study, easy to replicate and nonetheless, unique. I have some suggestion to improve clarity of your work and one main question that highlight below. Please check:

Abstract: Please add effect size in the abstract

Keywords: Please avoid using the same words that are already present in the title.

Introduction:

L42 - I presume that authors want to say: HSI have been associated with lower/reduced hamstring strength?!

L53-54 - please check the writing. Velocity two times does seem the right way to explain this type of training

L63-65 - the following study seems accurate to be referenced in this paragraph:

Antunes, J.P., Oliveira, R., Reis, V.M., Romero, F., Moutão, J., Brito, J.P. (2022). Comparison between Olympic Weightlifting Lifts and Derivatives for External Load and Fatigue Monitoring. Healthcare, 10, 2499. https://doi.org/10.3390/healthcare10122499

Materials and Methods:

L79 - What about the experience in 2 exercises analysed in the study?

L89 - I imagine that these 2 occasions included 1 day for deadlift and 1 day for hip thrust, but this was not explained. Please clarify

L98 - Please avoid writing in the first person. Please revise all text.

Statistical analysis:

- a sample power calculation or sample size calculation should be presented.

L135 – “were” appear twice and I believe that should be replaced by “was” in both times.  

L138-139 – 2 (sex) × 15 (load) - this part should be better explained in the testing procedures with references because, it is only clear when we observe table 1. Please check.

After that another question arises which is how did authors control fatigue? For instance, it is expected that after 4/5 sets, participants are more fatigated than before. This can influence results.

L142 – “<0.06” - for better clarity, it will be better providing a range interval here.

L181 - This ratio was not mentioned in methods. Please provide more details about this calculation. Also, in the statistical analysis section.

Discussion: Please start discussion by explaining the aims of the study.

L243 - Please provide more details on this statement. What do you mean?

References - References are not following Journal's guidelines.

Thank you so much for your consideration.

Round 2

Reviewer 1 Report

Overall comment:

The corrections made by the Authors make the manuscript improve the clarity of the manuscript and make it suitable for publication.

Specific comments:

LINE 76:               please modify “proposing” to “proposed”

LINE 153:             please modify “testing” to “tested”

LINE 155:             please modify “add” to “adding”

LINE 259:             Please modify “Despite this men” to “Despite this, men”

LINE 273:             please modify “their” to “tested”

LINE 292-294:     unclear, please rephrase

Reviewer 2 Report

Major questions

- Just a comment: several modifications were performed and were not marked in the new version. For example, the manuscript increased from 36 to 58 citations).

- 3.3 Stability in the load-velocity relationship regardless of individual relative strength: the results of these outcomes were withdrawn in the new version of the manuscript. However, some of the old discussion continued in the current version (see below).

- Line 256: “...Furthermore, differences in fat-free mass (FFM) between sexes may partly account for the results observed in this study [37], in which women produced significantly higher velocities when normalized to FFM”. This phrase was in the old version and does not match the current results and current discussion. It is necessary to carry out an important revision of the content of the discussion of the results.

Minor questions

- Line 78: “Recent studies have shown that the LV relationship is stable...”; Wouldn't it be FV relationship?

- Line 294: “LV profile as they have shown...” Wouldn't it be FV profile?

- References: a) standardize manuscript titles (are the initial letters capitalized or not? standardize); b) standardize journal name: J strength Cond Res (initial letters capitalized)
